# Molecular Diversity of *Giardia duodenalis*, *Cryptosporidium* spp., and *Blastocystis* sp. in Symptomatic and Asymptomatic Schoolchildren in Zambézia Province (Mozambique)

**DOI:** 10.3390/pathogens10030255

**Published:** 2021-02-24

**Authors:** Aly S. Muadica, Pamela C. Köster, Alejandro Dashti, Begoña Bailo, Marta Hernández-de-Mingo, Sooria Balasegaram, David Carmena

**Affiliations:** 1Parasitology Reference and Research Laboratory, National Centre for Microbiology, Health Institute Carlos III, Majadahonda, 28220 Madrid, Spain; amuandica@unilicungo.ac.mz (A.S.M.); pamkoste@ucm.es (P.C.K.); adashti@ucm.es (A.D.); begobb@isciii.es (B.B.); martaher1@hotmail.com (M.H.-d.-M.); 2Departamento de Ciências e Tecnologia, Universidade Licungo, 106 Quelimane, Zambézia, Mozambique; 3Field Epidemiology Services, National Infection Service, Public Health England, London SE1 8UG, UK; Sooria.Balasegaram@phe.gov.uk

**Keywords:** *Giardia*, *Cryptosporidium*, *Blastocystis*, enteric parasites, children, diarrhoea, PCR, molecular epidemiology, genotyping, Mozambique

## Abstract

Infections by the protist enteroparasites *Giardia duodenalis*, *Cryptosporidium* spp., and, to a much lesser extent, *Blastocystis* sp. are common causes of childhood diarrhoea in low-income countries. This molecular epidemiological study assesses the frequency and molecular diversity of these pathogens in faecal samples from asymptomatic schoolchildren (*n* = 807) and symptomatic children seeking medical attention (*n* = 286) in Zambézia province, Mozambique. Detection and molecular characterisation of pathogens was conducted by polymerase chain reaction (PCR)-based methods coupled with Sanger sequencing. *Giardia duodenalis* was the most prevalent enteric parasite found [41.7%, 95% confidence interval (CI): 38.8–44.7%], followed by *Blastocystis* sp. (14.1%, 95% CI: 12.1–16.3%), and *Cryptosporidium* spp. (1.6%, 95% CI: 0.9–2.5%). Sequence analyses revealed the presence of assemblages A (7.0%, 3/43) and B (88.4%, 38/43) within *G. duodenalis*-positive children. Four *Cryptosporidium* species were detected, including *C. hominis* (30.8%; 4/13), *C. parvum* (30.8%, 4/13), *C. felis* (30.8%, 4/13), and *C. viatorum* (7.6%, 1/13). Four *Blastocystis* subtypes were also identified including ST1 (22.7%; 35/154), ST2 (22.7%; 35/154), ST3 (45.5%; 70/154), and ST4 (9.1%; 14/154). Most of the genotyped samples were from asymptomatic children. This is the first report of *C. viatorum* and *Blastocystis* ST4 in Mozambique. Molecular data indicate that anthropic and zoonotic transmission (the latter at an unknown rate) are important spread pathways of diarrhoea-causing pathogens in Mozambique.

## 1. Introduction

Diarrhoea has long been recognized as a major cause of child morbidity and mortality globally [1]. The main diarrhoea-causing agents include viral (adenovirus, rotavirus), bacterial (*Campylobacter*, enterotoxigenic *Escherichia coli*, *Shigella*) and parasitic (*Giardia duodenalis*, *Cryptosporidium* spp.) pathogens. Recent data by the Global Burden of Diseases, Injuries, and Risk Factors Study (GBD) revealed that diarrhoea was the fifth leading cause of death among children younger than 5 years (446,000 deaths) only in 2016 [2]. In this group of age, diarrhoea causes more deaths than acquired immunodeficiency syndrome (AIDS), malaria, and measles combined [3]. Additionally, near 90% of diarrhoea-associated deaths are attributable to unsafe water, inadequate sanitation, and insufficient hygiene [4]. There is, therefore, a clear link between diarrhoea and poverty, particularly in deprived areas of low-income countries [5].

The protozoa *G. duodenalis* and *Cryptosporidium* spp. and, to a lesser extent, the Stramenopile *Blastocystis* sp. are among the most important diarrheal pathogens of parasitic nature affecting humans [6]. Indeed, there are more than 200 million cases of symptomatic giardiosis worldwide each year only in developing countries [7]. The Global Enteric Multicenter Study (GEMS) has demonstrated that *Cryptosporidium* is second only to rotavirus in causing morbidity and mortality in young children living in South East Asian and sub-Saharan African countries [8]. *Blastocystis* sp. is the most prevalent intestinal microbial eukaryote colonising/infecting the human gut. Although its clinical significance remains controversial, *Blastocystis* carriage has been linked with intestinal (diarrhoea, irritable bowel syndrome) and extra-intestinal (urticarial) disorders [9]. In addition, childhood cryptosporidiosis and giardiosis frequently lead to stunted growth and cognitive impairment, particularly in poor-resource areas [10,11].

As with other enteric protists, *G. duodenalis*, *Cryptosporidium* spp., and *Blastocystis* sp. are transmitted via the faecal-oral route through diverse pathways including person-to-person or animal-to-person direct contact and consumption of contaminated water or food. Indeed, waterborne and foodborne illnesses constitute a major area of concern [12,13]. Molecular epidemiological studies involving human, animal, and environmental samples are essential to determine the genetic diversity of these pathogens in a given host or geographical area, and to ascertain the exact contribution of the above-mentioned transmission routes to human infections.

*Giardia duodenalis* (the only *Giardia* species able to infect humans) is currently regarded as a multi-species complex comprising eight (A to H) distinct assemblages, of which zoonotic assemblages A and B (particularly the latter) are the most prevalent in humans [14]. The genus *Cryptosporidium* comprises at least 40 valid species and a similar number of genotypes of unclear taxonomic status, of which more than 20 have been reported to cause human infections. Only two species, *Cryptosporidium hominis* and *C. parvum*, account for up to 90% of the human cases of cryptosporidiosis documented globally [15]. Similarly, 22 distinct subtypes (ST) have been identified within *Blastocystis* sp. to date (ST1-17, ST21, ST23-26), of which ST1–9 and ST12 have been reported in humans [16].

The epidemiology of giardiosis, cryptosporidiosis, and blastocystosis in African countries is poorly understood [17,18,19,20]. Prevalence estimations have been frequently based on low-sensitive (e.g., microscopy) methods, were restricted to certain populations and geographical areas or are in need of update. When compared, polymerase chain reaction (PCR) methods outperformed microscopy in terms of sensitivity and range of parasite species detected [21]. Although recent large-scale epidemiological surveys such as GEMS or the Etiology, Risk Factors, and Interactions of Enteric Infections and Malnutrition and the Consequences for Child Health (MAL-ED) Study have considerably improved our knowledge of certain species (particularly *Cryptosporidium* spp.), they did not consider others (e.g., *Blastocystis*) and did not conduct genotyping analyses on positive samples [8,22]. In Mozambique, GEMS-derived samples from young children positive for *G. duodenalis* and *Cryptosporidium* spp. in the Manhiça District (Maputo province) have been recently genotyped and sub-genotyped [23,24]. Earlier studies have molecularly characterised much smaller numbers of *Giardia*- and *Cryptosporidium*-positive isolates in diarrhoeal individuals and patients with human immunodeficiency virus (HIV) and/or tuberculosis in Maputo and Gaza provinces, respectively [25,26]. However, the current epidemiological situation of human blastocystosis in the country is completely unknown. This study aims to investigate the frequency and molecular diversity of *G. duodenalis*, *Cryptosporidium* spp., and *Blastocystis* sp. in asymptomatic and symptomatic children in the Zambézia province in the central coastal region of Mozambique, an area where no studies on the presence of intestinal protist parasites have been previously conducted in human or animal populations.

## 2. Results

### 2.1. Occurrence of Enteric Parasites

A total of 1093 children aged 3–14 years participated in the present study, of which 807 children were enrolled from 18 schools and resided in 66 neighbourhoods in 10 districts of the Zambézia province. The other 286 children were enrolled from six primary healthcare centres and a hospital clinic and resided in 37 different neighbourhoods in six districts. Overall, *G. duodenalis* was the most prevalent enteric parasite found [41.7%, 95% confidence interval (CI): 38.8–44.7%], followed by *Blastocystis* sp. (14.1%, 95% CI: 12.1–16.3%), and *Cryptosporidium* spp. (1.6%, 95% CI: 0.9–2.5%). The prevalence rates of these pathogens in each participating school and healthcare centre are summarized in Table 1. Estimates did not consider the clustered nature of the data, as this task was thoroughly conducted elsewhere [27].

### 2.2. Prevalence and Molecular Characterization of G. duodenalis

A total of 456 DNA isolates tested positive for *G. duodenalis* by real-time polymerase chain reaction (qPCR, 336 and 120 in asymptomatic and symptomatic children, respectively). Generated cycle threshold (Ct) values had median values of 31.6 (range: 18.0–42.1) in asymptomatic children, and of 31.2 (range: 19.7–41.4) in symptomatic children. Overall, 45.8% (209/456) had qPCR values ≥32, and 39.7% (181/456) had qPCR Ct values <30. Based on previous molecular studies conducted by our research team using this very same method in human populations from other African countries including Mozambique [23,28,29], only DNA isolates with qPCR Ct values <32 (*n* = 247) were assessed for genotyping and sub-genotyping purposes in order to optimise available resources.

Assemblage/sub-assemblage assignment was conducted by direct comparison of the sequencing results obtained at the three loci (*gdh*, *bg*, and *tpi*) investigated. Sequences presenting double peak positions that could not be assigned unequivocally to a given assemblage/sub-assemblage were reported as ambiguous sequences. Out of the 247 DNA isolates investigated, 15.0% (37/247), 10.1% (25/247), and 6.1% (15/247) yielded amplicons at the *gdh*, *bg*, and *tpi* loci, respectively (Table 2). Overall, 17.4% (43/247) were amplified at least at a single locus, whereas multi-locus genotyping data at the three loci were available for 4% (10/247) of them. Most (88.4%, 38/43) of the isolates successfully amplified at any of the three markers assessed had qPCR Ct values <30. Sequence analyses revealed the presence of assemblages A (7.0%, 3/43) and B (88.4%, 38/43). Two additional sequences (4.6%, 2/43) corresponded to mixed A+B infections. No infections caused by host-restricted canine (C, D), feline (F), or ruminant (E) assemblages were detected. All genotyped isolates except one were obtained in asymptomatic children.

All three A sequences were assigned to the sub-assemblage AII of the parasite. Out of the 40 B sequences, 12.5% (5/40) were identified as sub-assemblage BIII, 20.0% (8/40) as sub-assemblage BIV, 52.5% (21/40) as ambiguous BIII/BIV sequences, and the remaining 15.0% (6/40) were only genotyped at the assemblage level.

The diversity, frequency, and main features of the *G. duodenalis* sequences generated at the *gdh* locus are shown in Appendix A. Briefly, all four AII sequences were identical to reference sequence L40510. In contrast, a much higher level of genetic diversity was observed within the 34 sequences assigned to assemblage B at this locus. Indeed, the five sequences identified as BIII differed by 1–9 single nucleotide polymorphisms (SNPs) from reference sequence AF069059, most of them associated to heterozygous C/T peaks at positions 99, 147, 150, 309, and 336. The nine sequences unambiguously assigned to BIV differed by 2–8 SNPs from reference sequence L40508. Six of them presented nucleotide substitutions (mainly C↔T transitions) at positions 183, 387, 396, and 423, but not ambiguous positions in the form of double peaks. SNPs present in the three remaining BIV sequences combined mutations and heterozygous positions at different proportions. Remarkably, virtually all ambiguous BIII/BIV sequences different among them and by 5–15 SNPs from reference sequence L40508. Most of these SNPs involved heterozygous C/T (and, to a lesser extent, A/G) peaks. In contrast, SNPs associated to transition C↔T or A↔G mutations were rare or non-existent. Some of these ambiguous BIII/BIV sequences presented clear double peaks at defined positions specific for BIII (e.g., 99, 147, 150, 309, and 336) and BIV (e.g., 183, 387, 396, and 423) sequences, suggesting the occurrence (at an unknown rate) of true BIII+BIV intra-assemblage mixed infections. Several heterozygous positions within BIII and BIII/BIV (particularly the latter) sequences were potentially associated to amino acid change in the polypeptidic chain.

The diversity, frequency, and main features of the 25 *G. duodenalis* sequences generated at the *bg* locus are summarized in Appendix A. The only sequence assigned to AII was identical to reference sequence AY072723. The other 24 sequences, all belonging to the assemblage B of *G. duodenalis*, presented a comparatively lower degree of genetic diversity than their counterparts at the *gdh* locus. Of them, four sequences were identical to reference sequence AY072727, whereas the remaining 20 sequences differed from it by 1–5 SNPs. Variations involving transitional C↔T or A↔G mutations and double peaks tended to accumulate at positions 183, 309, 519, and 565 of AY072727; some of them (including a transversion C/A mutation) were involved in amino acid substitutions at the protein level.

The diversity, frequency, and main features of the 15 *G. duodenalis* sequences generated at the *tpi* locus were summarized in Appendix A. Out of the three sequences identified as AII, two of them varied by 1–2 SNPs (including a transversion C/G mutation at position 287) from reference sequence U57897. The third AII sequence lacked sufficient quality to accurately determine the presence of potential SNPs. The 10 sequences characterised as BIII differed by 1–8 SNPs from reference sequence AF069561. Six of them included only transitional C↔T or A↔G mutations, whereas the remaining four had several heterozygous positions. Detected SNPs tended to accumulate at positions 34, 108, and 141 of AF069561, some of them involved in amino acid substitutions in the polypeptidic chain. Two isolates corresponded to ambiguous BIII/BIV sequences differing by seven SNPs from reference sequence AF069560. Most of these SNPs were the result of transitional C/T or A/G mutations, one of them (T57C) involving an amino acid chain at the protein level. No transversion mutations were detected within sequences generated at the *tpi* locus.

The evolutionary relationships among the *G. duodenalis* sequences generated at the *gdh* locus in the present study were shown in Figure 1. Sequences of human origin generated by our research team in previous studies conducted in geographical areas of high (Ethiopia, Angola, Brazil, and Iran) and low (Spain) endemicity were also included in the analysis for comparative purposes. Assemblage A sequences grouped together in well-defined clusters with appropriate reference sequences. Although assemblage B sequences also formed a well-supported clade (88% of bootstrap), sub-assemblage BIII and BIV sequences could not be segregated in independent clusters. Phylogenetic trees generated at the *bg* (Appendix A) and *tpi* (Appendix A) loci seem to corroborate this finding.

### 2.3. Prevalence and Molecular Characterization of Cryptosporidium *spp.*

Analyses of the 13 *ssu* rDNA sequences generated in the present study revealed the presence of four *Cryptosporidium* species circulating in the paediatric population under study, including *C. hominis* (30.8%; 4/13), *C. parvum* (30.8%, 4/13), *C. felis* (30.8%, 4/13), and *C. viatorum* (7.6%, 1/13) (Table 3). Besides a case of cryptosporidiosis by *C. parvum* detected in a child with diarrhoea, the remaining cases were identified in asymptomatic children. Out of the four sequences characterised as *C. hominis*, two of them were identical to reference sequence AF108865, with the remaining two differing from it by one to two SNPs including a nucleotide deletion and ambiguous positions in the form of double peaks. Three of the sequences assigned to *C. parvum* corresponded to the “bovine genotype” of the parasite, characterised by a four-nucleotide (TAAT) deletion involving positions 686_689 of reference sequence AF112571. These three sequences differed among them by 1–2 SNPs affecting positions 795, 837, and/or 892 of reference sequence AF112571. A fourth *C. parvum* sequence was not investigated further due to insufficient quality to accurately determine the presence of potential SNPs. All *C. hominis* and *C. parvum* isolates failed to be amplified at the *gp60* locus despite repeated attempts to do so. Therefore, their family subtypes remain unknown.

Three of the *C. felis* sequences were identical among them but differed from reference sequence AF108862 by three SNPs including a deletion, an insertion, and a substitution (Table 3). The fourth *C. felis* sequence had an additional T insertion in position 826 of reference sequence AF108862. The only sequence assigned to *C. viatorum* was identical to positions 290–762 of reference sequence KX174309. Sequence analysis at the *gp60* locus revealed that this isolate belonged to the subtype XVaA3a of the parasite, showing 100% identity to positions 1–870 of reference sequence KP115936.

The genetic relationships among *ssu* rRNA gene sequences generated in the present study, as inferred by a neighbor-joining analysis, were shown in Figure 2. All *Cryptosporidium* sequences clustered together (monophyletic groups) with different well-supported clades (58–99% of bootstrap) corresponding to appropriate reference sequences for *Cryptosporidium* species.

### 2.4. Prevalence and Molecular Characterization of Blastocystis *sp.*

Out of the 227 isolates that yielded amplicons of the expected size for *Blastocystis* sp. by PCR, 67.8% (154/227) were successfully subtyped at the *ssu* rRNA (barcode region) gene of this protist. The remaining 73 isolates produced unreadable or poor-quality sequences usually associated to faint bands on agarose gels. All samples that could not be confirmed by Sanger sequencing were presumptively considered as *Blastocystis*-negative. Overall, 92.9% (143/154) of the subtyped isolates were obtained in asymptomatic children, and 7.1% (11/154) in children with gastrointestinal symptoms. Sequence analyses allowed the identification of four *Blastocystis* subtypes (ST) circulating in the surveyed paediatric population, including ST1 (22.7%; 35/154), ST2 (22.7%; 35/154), ST3 (45.5%; 70/154), and ST4 (9.1%; 14/154) (Figure 1). Neither mixed infection involving different STs of the parasite nor infections caused by animal-specific ST10-ST17, ST21 or ST23-26 were identified. A considerable genetic diversity was observed within ST2 (four different alleles, alone or in combination) and ST3 (six different alleles, alone or in combination). Allele 4 was the most prevalent within ST1 (77.1%, 27/35), allele 12 within ST2 (34.3%, 12/35), and allele 36 within ST3 (54.3%, 38/70). In contrast, allele 42 was the only genetic variant identified within ST4. Several isolates were only genotyped at the subtype level due to insufficient quality sequence data for allele calling (Figure 3).

No obvious clusters of the parasite’s species/genotypes were found according to the period of sampling or the sex, age group, school of origin or clinical status of the infected children investigated in the present study.

## 3. Discussion

The molecular diversity of *G. duodenalis* and *Cryptosporidium* spp. in Mozambique has been thoroughly investigated in children younger than five years of age with and without diarrhoea recruited under the GEMS umbrella in the Manhiça district (Maputo province) in two recent studies [23,24]. The present survey expands our current knowledge on the epidemiology of these two protozoan pathogens in the country, exploring their occurrence and genetic diversity in asymptomatic schoolchildren and children with clinical manifestations in Zambézia province. This is also the first report describing the molecular variability of *Blastocystis* sp. in Mozambique. Of note, our research group has previously investigated in this very same paediatric populations the occurrence of the microsporidia *Enterocytozoon bieneusi* and conducted risk association analyses for intestinal parasites [27,30].

In Mozambique, *G. duodenalis* has been described at infection rates of 1–6% by conventional microscopy in paediatric and clinical populations in Maputo province [31,32,33], and of 8–37% by PCR in patients with HIV and/or tuberculosis in Gaza province and people living in a high endemic area of Sofala province [21,25]. Using enzyme-linked immunosorbent assay (ELISA), *G. duodenalis* was identified in 10–50% of children with clinical manifestations in different provinces of the country [34,35,36]. The *G. duodenalis* crude prevalence rate found here (42%) is in the upper range limit of those reported in the aforementioned surveys.

*Giardia duodenalis* genotyping data in Mozambique are far scarcer. In a preliminary study involving a limited number of isolates, *G. duodenalis* sub-assemblages AII and BIV were found at equal proportions in patients with HIV and/or tuberculosis in Gaza province [25]. A much larger molecular survey involving 222 well-characterized *G. duodenalis* isolates obtained during the GEMS project revealed that assemblage B caused 9 out of 10 infections in young children in Maputo province [23]. That study demonstrated that the occurrence of diarrhoea was not linked to a given assemblage of the parasite. Additionally, the elevated level of genetic variability found within assemblage B sequences did not allow the correct differentiation between sub-assemblage BIII and BIV sequences that were, therefore, tentatively identified as ambiguous BIII/BIV results and represented 59% (132/222) of the total sequences analysed. Molecular data showed in the present study were remarkably similar, with assemblages A and B being identified in 7% and 88% of the infections, respectively, and ambiguous BIII/BIV sequences accounting for 49% (21/43) of them. The absence of animal-specific C–F assemblages in both surveys suggest that livestock and companion animals may play a secondary role as sources of human giardiosis, and that most human infections should be of anthropic origin. Taken together, these results indicate that the epidemiology of human giardiosis is very similar in Mozambican regions as separated from each other as Maputo and Zambézia.

*Cryptosporidium* infections in Mozambique have been previously documented at rates ranging from 6–38% by ELISA in diarrhoeic children in Maputo province [34,35], at 6% in HIV-positive individuals in the same province [33], and of 12% in general population at national scale [36]. Rates lower than 10% have been reported by PCR in patients with HIV and/or tuberculosis in Gaza province and diarrhoeic patients in Maputo province [25,26]. A much lower crude prevalence of 1.6% has been reported in the present study in mostly asymptomatic children. The marked discrepancies observed in *Cryptosporidium* prevalence rates among these studies may be associated with differences not only in the nature of the surveyed populations, but also in the performance of the diagnostic methods used.

Knowledge on the molecular diversity of *Cryptosporidium* sp. in Mozambique is very limited. Early studies revealed the presence of three *Cryptosporidium* species circulating in Mozambican human populations, namely *C. hominis*, *C. parvum*, and *C. felis*. Furthermore, *gp60* subtypes IbA10G2 and IdA22 have been described in patients with HIV and/or tuberculosis [25], and IA23R3, IIcA5G3, and IIeA12G1 in children and adults with diarrhoea [26,37]. More recently, a large panel (*n* = 191) of *Cryptosporidium*-positive faecal samples from young children collected during the GEMS project in the Maputo province revealed the predominance of *C. hominis* (73%) over *C. parvum* (23%) and *C. meleagridis* (4%). Both *C. hominis* and *C. parvum* were more prevalently found in diarrhoeal children than in non-diarrhoeal children. In that survey, a high intra-species genetic variability was observed within *C. hominis* (subtype families Ia, Ib, Id, Ie, and If) and *C. parvum* (subtype families IIb, IIc, IIe, and IIi), but not within *C. meleagridis* (subtype family IIIb). In contrast, in the present study *C. hominis*, *C. parvum*, and *C. felis* were found at equal (31%) proportions mostly in asymptomatic children, whereas the presence of *C. viatorum* subtype XVaA3a represents the first report of this *Cryptosporidium* species in Mozambique. Unfortunately, sequences identified as *C. hominis* or *C. parvum* at the *ssu* rRNA gene did not yield amplicons at the *gp60* locus, so their subtypes remain unknown.

Of interest, three out of four sequences identified as *C. parvum* were associated to the “bovine genotype” of the parasite, which is characterised by a four-base deletion TAAT at positions 686 to 689 of reference sequence AF112571. Indeed, some authors have proposed that this genetic variant should be considered as an independent species named *C. pestis* [38]. The high proportion of infections due to the “bovine genotype” of *C. parvum* and *C. felis* (a *Cryptosporidium* species adapted to infect cats and other felids) reveals that a significant number of cryptosporidiosis cases were the result of zoonotic events through direct contact with infected animals or indirectly through consumption of contaminated water or food with their faecal material. Of interest, *C. felis* has been previously reported in different human populations in other African countries including Ethiopia, Nigeria, and Kenya [39,40,41]. The presence of *C. viatorum* is also relevant. This *Cryptosporidium* species was initially thought to be a human-specific species [42], but recent epidemiological surveys have demonstrated its presence in rodents from Australia and China and may have, therefore, zoonotic potential [43,44]. In Africa, *C. viatorum* has been reported in primarily asymptomatic children, diarrhoeic patients and HIV+ individuals in Ethiopia [28,39,45], and in the adult population in Kenya [45].

A major contribution of this study is the first thorough description of the molecular diversity of *Blastocystis* sp. conducted in Mozambique to date. The frequency and diversity of the three main STs detected (ST1: 23%, ST2: 23%, ST3: 45%) were in line with those previously published in other African countries such as Angola [29], Ivory Coast [46], and Madagascar [47], among others. In contrast, ST4 was identified at a much lower rate of 9%. Remarkably, most human cases of blastocystosis by ST4 have been documented in Europe [48]. This geographically restricted pattern of ST4, together with its primarily clonal structure, has been interpreted by some authors as the result of a recent entry into the human population, very likely from rodents [49]. In line with these findings, *Blastocystis* ST4 has only been detected in a few African countries at carriage rates of 12–14% in Liberia and Nigeria [48], and of 2% in Senegal and Tunisia [50,51]. Furthermore, ST4 has been proposed as a more virulent *Blastocystis* strain, being linked with diarrhoeic patients in Denmark and Spain [52,53], and with irritable bowel syndrome and chronic diarrhoea in patients in Italy [54]. This does not seem to be the case of the present study, where all the *Blastocystis* isolates characterised as ST4 were identified in apparently healthy children. Absence of STs rarely found in humans (ST5-ST9) or thought to be present only in non-human animal species (ST10-ST17, ST21, ST23-ST26) seem to indicate that transmission of blastocystosis in Zambézia province is mainly of anthropic origin.

This study benefits from the high number of participating children and number of samples analysed, which allowed for a robust estimate of prevalences. Genetic data were strengthened by the adoption of a multi-locus genotyping scheme for the molecular characterization of samples positive to *G. duodenalis* and *Cryptosporidium* spp. However, the survey also presents some limitations that must be taken into consideration when interpreting the obtained results. Perhaps the most relevant is the long period (up to three months) that elapsed between sample collection and sample processing and analysis. During this time, collected stool samples were kept at room temperature in commercial devices intended to preserve the specimens and allow their use in downstream molecular assays. Despite this effort, suboptimal conservation of stool samples may have negatively affected the quality of the purified genomic DNA. This may explain the low proportion of *G. duodenalis*- and *Cryptosporidium*-positive isolates that were successfully amplified at their respective genotyping loci (*gdh*, *bg*, and *tpi* for *G. duodenalis*, *gp60* for *Cryptosporidium*). It is very likely that extraction and purification of DNA from fresh specimens would have significantly improved the genotyping data presented here. Finally, this study focused on human populations only. No attempts were conducted to analyse samples from animal and environmental (e.g., drinking water) sources, so the picture of the epidemiology of giardiosis, cryptosporidiosis and blastocystosis in this geographical area remains incomplete. This task must be accomplished in future molecular surveys.

## 4. Materials and Methods

### 4.1. Study Area and Stool Sample Collection

A prospective cross-sectional molecular epidemiological study of diarrhoea-causing enteric parasites including the protozoan *G. duodenalis* and *Cryptosporidium* spp. and the stramenopile *Blastocystis* sp., was conducted with children aged 3–14 from 10 of the 22 districts of Zambézia province, central Mozambique, between October 2017 and February 2019. Stool samples were collected from participating schoolchildren attending 18 primary schools or children seeking medical attention at seven primary healthcare centres (Figure 4). In school settings (range: 35–2111; mean: 651 schoolchildren) informative meetings were held for interested families. Schoolchildren volunteering to participate were given sampling kits to obtain stool samples during school attendance. In primary health clinics, children with gastrointestinal complaints (chronic or acute diarrhoea, bloating, abdominal pain) were invited to participate in the survey. Samples were collected by members of the research team at scheduled times and an aliquot (2–3 g) transferred to REAL Minisystem devices (Durviz, Valencia, Spain) for stool sample conservation and concentration. Preserved samples were maintained at room temperature up to three months before being transported to the Spanish National Centre for Microbiology (Majadahonda, Spain) for processing and analysis. Inclusion and exclusion criteria to participate in the study, school features, and sampling procedures as well as the detailed analysis of potential associations linked with enteric parasite infections in the paediatric populations surveyed here were thoroughly described elsewhere (Muadica et al., 2020) [27].

### 4.2. DNA Extraction and Purification

Genomic DNA was isolated from about 200 mg of each faecal specimen by using the QIAamp DNA Stool Mini Kit (Qiagen, Hilden, Germany) according to the manufacturer’s instructions, except that samples mixed with InhibitEX buffer were incubated for 10 min at 95 °C. Extracted and purified DNA samples (200 μL) were kept at −20 °C until further molecular analysis. A water extraction control was included in each sample batch processed.

### 4.3. Molecular Detection and Characterization of Giardia duodenalis

*Giardia duodenalis* DNA was detected by qPCR amplification of a 62 bp-fragment of the small subunit ribosomal RNA (*ssu* rRNA) gene of the parasite [55]. Amplification reactions (25 μL) consisted of 3 μL of template DNA, 0.5 μM of primers Gd-80F and Gd-127R, 0.4 μM of probe (Additional file 1: Appendix A), and 12.5 μL TaqMan^®^ Gene Expression Master Mix (Applied Biosystems, CA, USA). Detection of parasitic DNA was performed on a Corbett Rotor Gene™ 6000 real-time PCR system (Qiagen) using an amplification protocol consisting on an initial hold step of 2 min at 55 °C and 15 min at 95 °C followed by 45 cycles of 15 s at 95 °C and 1 min at 60 °C. Water (no-template) and genomic DNA (positive) controls were included in each PCR run.

*Giardia duodenalis* isolates with a qPCR-positive result were re-assessed by sequence-based multi-locus genotyping of the genes encoding for the glutamate dehydrogenase (*gdh*), ß-giardin (*bg*) and triose phosphate isomerase (*tpi*) proteins of the parasite. A semi-nested PCR was used to amplify a 432-bp fragment of the *gdh* gene [56]. PCR reaction mixtures (25 μL) included 5 μL of template DNA and 0.5 μM of the primer pairs GDHeF/GDHiR in the primary reaction and GDHiF/GDHiR in the secondary reaction (Appendix A). Both amplification protocols consisted of an initial denaturation step at 95 °C for 3 min, followed by 35 cycles of 95 °C for 30 s, 55 °C for 30 s and 72 °C for 1 min, with a final extension of 72 °C for 7 min. A nested PCR was used to amplify a 511 bp fragment of the bg gene [57]. PCR reaction mixtures (25 μL) consisted of 3 μL of template DNA and 0.4 μM of the primers sets G7_F/G759_R in the primary reaction and G99_F/G609_R in the secondary reaction (Appendix A). The primary PCR reaction was carried out with the following amplification conditions: one step of 95 °C for 7 min, followed by 35 cycles of 95 °C for 30 s, 65 °C for 30 s, and 72 °C for 1 min with a final extension of 72 °C for 7 min. The conditions for the secondary PCR were identical to the primary PCR except that the annealing temperature was 55 °C. Finally, a nested PCR was used to amplify a 530 bp-fragment of the *tpi* gene [58]. PCR reaction mixtures (50 μL) included 2–2.5 μL of template DNA and 0.2 μM of the primer pairs AL3543/AL3546 in the primary reaction and AL3544/AL3545 in the secondary reaction (Appendix A). Both amplification protocols consisted of an initial denaturation step at 94 °C for 5 min, followed by 35 cycles of 94 °C for 45 s, 50 °C for 45 s and 72 °C for 1 min, with a final extension of 72 °C for 10 min.

### 4.4. Molecular Detection and Characterization of Cryptosporidium *spp.*

The presence of *Cryptosporidium* spp. was assessed using a nested PCR to amplify a 587-bp fragment of the *ssu* rRNA gene of the parasite [59]. Amplification reactions (50 μL) included 3 μL of DNA sample and 0.3 μM of the primer pairs CR-P1/CR-P2 in the primary reaction and CR-P3/CPB-DIAGR in the secondary reaction (Appendix A). Both PCR reactions were carried out as follows: one step of 94 °C for 3 min, followed by 35 cycles of 94 °C for 40 s, 50 °C for 40 s and 72 °C for 1 min, concluding with a final extension of 72 °C for 10 min.

Isolates identified as *C. hominis* or *C. parvum* by *ssu*-PCR (and Sanger sequencing, see below) were reanalysed at the 60 kDa glycoprotein (*gp60*) gene for subtyping purposes. Briefly, a nested PCR was conducted to amplify a 870 bp fragment of the *gp60* locus [60]. PCR reaction mixtures (50 μL) included 2–3 μL of template DNA and 0.3 μM of the primer pairs AL-3531/AL-3535 in the primary reaction and AL-3532/AL-3534 in the secondary reaction (Appendix A). The primary PCR reaction consisted of an initial denaturation step of 94 °C for 5 min, followed by 35 cycles of 94 °C for 45 s, 59 °C for 45 s, and 72 °C for 1 min with a final extension of 72 °C for 10 min. The conditions for the secondary PCR were identical to the primary PCR except that the annealing temperature was 50 °C. Similarly, samples identified as *C. viatorum* by *ssu*-PCR were reanalysed at the *gp60* locus using a nested PCR protocol specifically developed for this *Cryptosporidium* species [45]. This protocol amplifies a 950 bp fragment of the *gp60* gene. Amplification reactions (50 μL) included 1–2 μL of DNA sample and 0.25 μM of the primer pair CviatF2/CviatR5 in the primary reaction and 0.5 µM of the primer pair CviatF3/CviatR8 in the secondary reaction (Appendix A). Both PCR reactions were carried out as follows: one step of 95 °C for 4 min, followed by 35 cycles of 95 °C for 30 s, 58 °C for 30 s and 72 °C for 1 min, concluding with a final extension of 72 °C for 7 min.

### 4.5. Molecular Detection and Characterization of Blastocystis *sp.*

Identification of *Blastocystis* sp. was achieved by a direct PCR targeting the *ssu* rRNA gene of the parasite [61]. This protocol uses the pan-*Blastocystis*, barcode primers RD5 and BhRDr (Appendix A) to amplify a PCR product of 600 bp. Amplification reactions (25 μL) included 5 μL of template DNA and 0.5 μM of the primer set RD5/BhRDr. Amplification conditions consisted of one step of 95 °C for 3 min, followed by 30 cycles of 1 min each at 94, 59 and 72 °C, with an additional 2 min final extension at 72 °C.

All the direct, semi-nested, and nested PCR protocols described above were conducted on a 2720 thermal cycler (Applied Biosystems). Reaction mixes always included 2.5 units of MyTAQ™ DNA polymerase (Bioline GmbH, Luckenwalde, Germany), and 5× MyTAQ™ Reaction Buffer containing 5 mM dNTPs and 15 mM MgCl_2_. Laboratory-confirmed positive and negative DNA isolates for each parasitic species investigated were routinely used as controls and included in each round of PCR. PCR amplicons were visualized on 2% D5 agarose gels (Conda, Madrid, Spain) stained with Pronasafe nucleic acid staining solution (Conda). Positive-PCR products were directly sequenced in both directions using the internal primer set described above. DNA sequencing was conducted by capillary electrophoresis using the BigDye^®^ Terminator chemistry (Applied Biosystems) on an ABI PRISM 3130 automated DNA sequencer.

### 4.6. Sequence and Phylogenetic Analyses

Raw sequencing data in both forward and reverse directions were viewed using the Chromas Lite version 2.1 sequence analysis program (https://technelysium.com.au/wp/chromas/, accessed on 23 February 2021). The BLAST tool (http://blast.ncbi.nlm.nih.gov/Blast.cgi, accessed on 23 February 2021) was used to compare nucleotide sequences with sequences retrieved from the NCBI GenBank database. Generated DNA consensus sequences were aligned to appropriate reference sequences using the MEGA 6 software [62] to identify *G. duodenalis* assemblages/sub-assemblages and *Cryptosporidium* species. *Blastocystis* sequences were submitted at the *Blastocystis* 18S database (http://pubmlst.org/blastocystis/, accessed on 23 February 2021) for sub-type confirmation and allele identification. 

The evolutionary relationships among the identified *G. duodenalis* assemblages/sub-assemblages at the three loci investigated and the *Cryptosporidium* species found were inferred by a phylogenetic analysis using the neighbor-joining method in MEGA 6. Only sequences with unambiguous (no double peak) positions were used in the analyses. The evolutionary distances were computed using the Kimura 2-parameter method and modelled with a gamma distribution. The reliability of the phylogenetic analyses at each branch node was estimated by the bootstrap method using 1000 replications. Representative sequences of different *G. duodenalis* assemblages and sub-assemblages and *Cryptosporidium* species were retrieved from the NCBI database and included in the phylogenetic analyses for reference and comparative purposes.

Representative *G. duodenalis* sequences obtained in this study have been deposited in GenBank under accession numbers MW508361–MW508394 (*gdh* locus), MW508394–MW508410 (*bg* locus) and MW556751–MW556764 (*tpi* locus). Representative *Cryptosporidium* spp. sequences were deposited under accession numbers MW563962–MW563970 (*ssu* rRNA locus) and MW574004 (*gp60* locus). Representative *Blastocystis* sp. sequences were deposited under accession numbers MW564221–MW564233.

## 5. Conclusions

This PCR-based epidemiological study provides novel data on the molecular diversity of *G. duodenalis*, *Cryptosporidium* spp., and *Blastocystis* sp. in children with and without diarrhoea in Zambézia province, a Mozambican region where this information was completely lacking. Generated results complement and expand those obtained previously by the GEMS project in paediatric populations in Maputo province. A high intra-species molecular diversity was observed among the three parasites investigated, a finding compatible with an epidemiological scenario of high endemicity where infections and reinfections were common. No obvious differences in the distribution and frequency of parasite’ species/genotypes were observed between apparently healthy children and children with clinical manifestations, suggesting that virulence/pathogenicity was not associated to a given genetic variant. Transmission of giardiosis and blastocystosis was primarily of anthropic nature, but strong molecular evidence indicated that a significant number of cryptosporidiosis cases were the result of zoonotic events. Data presented here highlight the need to conduct new molecular surveys in animal and environmental (drinking water) samples to complete our understanding of the transmission dynamics of these protist species in Mozambique. Measures directed to improve access to safe drinking water, sanitation facilities, and personal hygiene (e.g., hand washing) practices would definitively help in minimizing the transmission of diarrhoea-causing pathogens in highly endemic areas in Mozambique.

## Figures and Tables

**Figure 1 pathogens-10-00255-f001:**
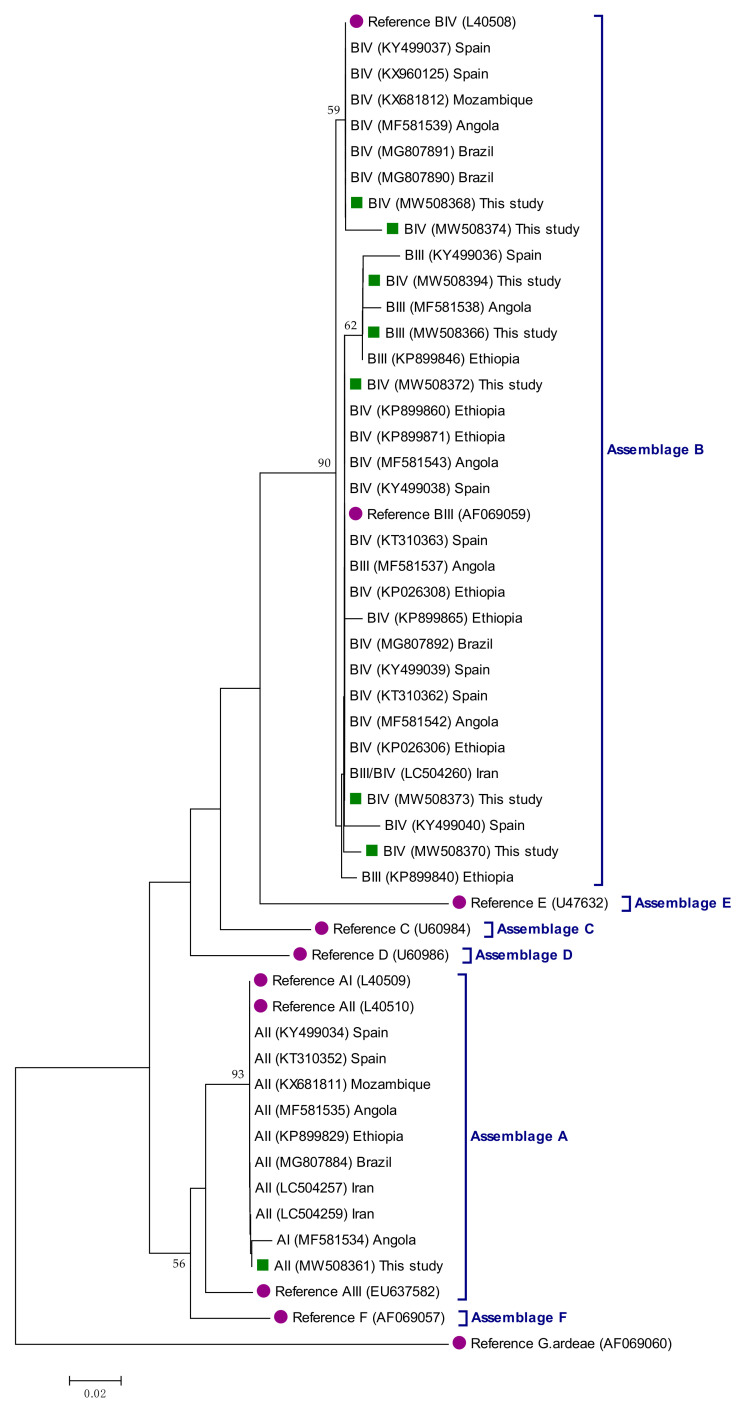
Phylogenetic relationships among *Giardia duodenalis* assemblages and sub-assemblages identified in infected symptomatic and asymptomatic children in the Zambézia province, Mozambique. The analysis was conducted by a neighbor-joining method of a 412-bp fragment (corresponding to position 79–490 of reference sequence L40508) of the *gdh* gene sequence. Genetic distances were calculated using the Kimura two-parameter model. Green filled squares represent sequences generated in the present study. Purple filled dots represent reference sequences. Bootstrap values lower than 50% are not displayed. *Giardia ardeae* was used as outgroup taxon to root the tree.

**Figure 2 pathogens-10-00255-f002:**
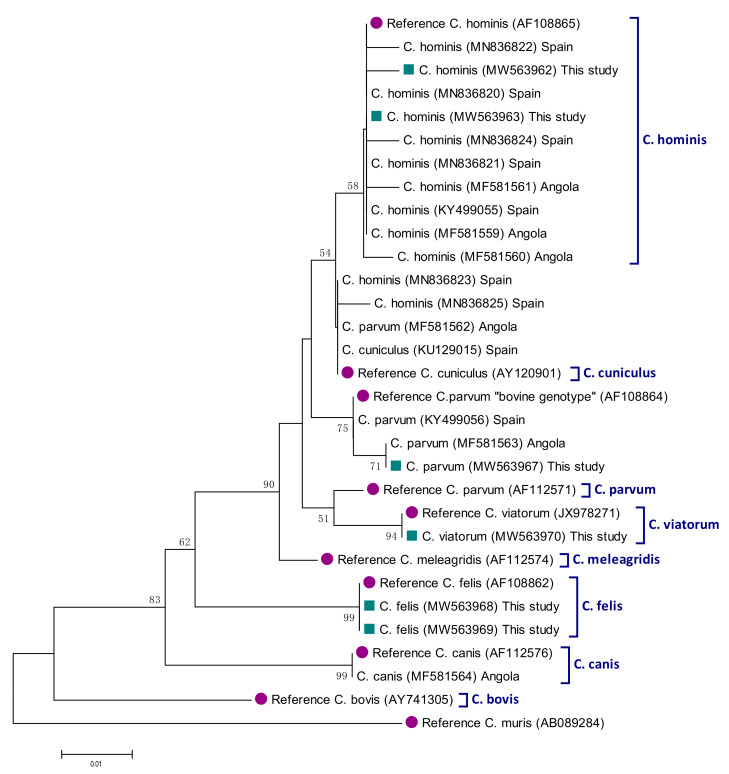
Phylogenetic relationships among *Cryptosporidium* species identified in infected SympTable 490. bp fragment (corresponding to position 538–1027 of reference sequence AF108865) of the *ssu* rRNA gene sequence. Genetic distances were calculated using the Kimura two-parameter model. Green filled squares represent sequences generated in the present study. Purple filled dots represent reference sequences. Bootstrap values lower than 50% are not displayed. *Cryptosporidium muris* was used as outgroup taxon to root the tree.

**Figure 3 pathogens-10-00255-f003:**
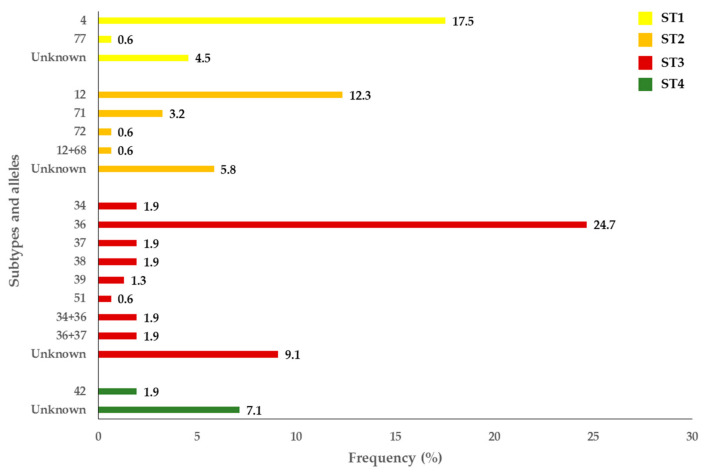
Diversity and frequency of *Blastocystis* subtypes and 18S alleles identified in the symptomatic and asymptomatic children in Zambézia province, Mozambique.

**Figure 4 pathogens-10-00255-f004:**
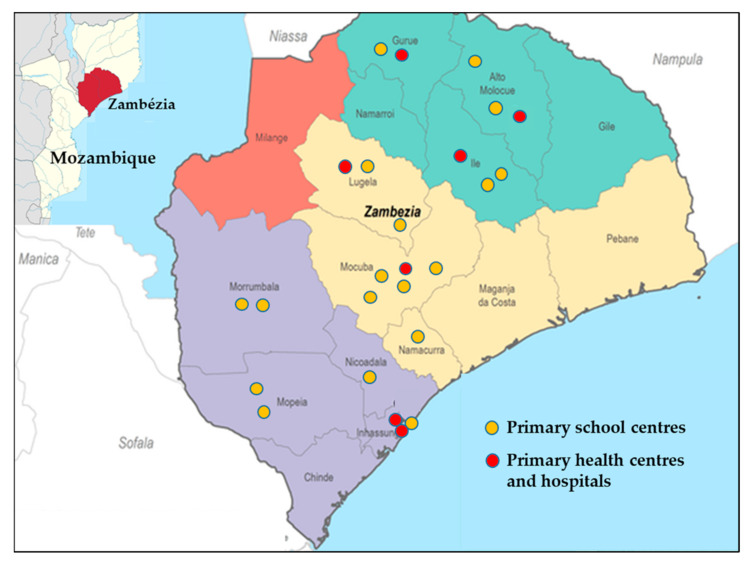
Map showing the geographical location of the Zambézia province in Mozambique (upper left corner) and the primary school and healthcare centres sampled in the present study.

**Table 1 pathogens-10-00255-t001:** Molecular-based prevalence rates of *Giardia duodenalis*, *Cryptosporidium* spp., and *Blastocystis* sp. in the surveyed paediatric population by school or medical centre of origin in the Zambézia province, Mozambique.

Centre	Children (*n*)	*Giardia duodenalis*		*Cryptosporidium* spp.		*Blastocystis* sp.	
		Positive (*n*)	Positive (%)	Positive (*n*)	Positive (%)	Positive (*n*)	Positive (%)
**School**							
1	44	23	52.3	1	2.3	1	2.3
2	50	13	26.0	0	0.0	23	46.0
3	22	9	40.9	0	0.0	0	0.0
4	22	12	54.5	0	0.0	0	0.0
5	24	11	45.8	0	0.0	0	0.0
6	31	13	41.9	0	0.0	1	3.2
7	88	36	40.9	4	4.5	9	10.2
8	60	8	13.3	2	3.3	0	0.0
9	47	23	48.9	0	0.0	0	0.0
10	49	20	40.8	0	0.0	1	2.0
11	47	13	27.7	0	0.0	4	8.5
12	50	12	24.0	0	0.0	4	8.0
13	50	25	50.0	1	2.0	19	38.0
14	30	9	30.0	0	0.0	0	0.0
15	30	5	16.7	0	0.0	0	0.0
16	40	16	40.0	0	0.0	0	0.0
17	75	46	61.3	1	1.3	47	62.7
18	48	42	87.5	2	4.2	34	70.8
Sub-total	807	336	41.6	11	1.4	143	17.7
**Clinic**							
1	50	32	64.0	3	6.0	2	4.0
2	25	11	44.0	0	0.0	0	0.0
3	15	4	26.7	0	0.0	2	13.3
4	52	28	53.8	0	0.0	4	7.7
5	42	10	23.8	2	4.8	2	4.8
6	29	4	13.8	1	3.4	1	3.4
7	73	31	42.5	0	0.0	0	0.0
Sub-total	286	120	42.0	6	2.1	11	3.8
Total	1093	456	41.7	17	1.6	154	14.1

**Table 2 pathogens-10-00255-t002:** Multilocus genotyping results of the *G. duodenalis*-positive children (*n* = 43) successfully genotyped at any of the three loci investigated in the Zambézia province, Mozambique.

**Sample ID**	**Ct Value in qPCR**	***gdh***	***bg***	***tpi***	**Assigned Genotype**
4	24.49	BIV	Negative	Negative	BIV
17	25.8	BIV	Negative	Negative	BIV
24 ^1^	31.6	BIV	Negative	Negative	BIV
57	25.4	BIV	Negative	Negative	BIV
62	20.9	AII	Negative	AII	AII
67	23.9	BIII/BIV	Negative	Negative	BIII/BIV
69	21.9	BIV	B	BIII	BIII/BIV
70	24.3	BIII/BIV	B	Negative	BIII/BIV
74	22.7	BIII	B	Negative	BIII
79	26.1	AII	Negative	Negative	AII
88	20.9	BIII/BIV	Negative	BIII	BIII/BIV
97	26.1	BIV	Negative	Negative	BIV
100	25.9	BIII/BIV	B	Negative	BIII/BIV
103	27.5	BIII/BIV	Negative	Negative	BIII/BIV
104	21.7	BIII/BIV	B	BIII	BIII/BIV
105	24.0	BIII/BIV	B	Negative	BIII/BIV
118	26.8	BIV	B	Negative	BIV
122	20.0	BIII/BIV	B	BIII/BIV	BIII/BIV
124	30.8	Negative	B	Negative	B
128	31.8	Negative	B	Negative	B
140	30.5	Negative	B	Negative	B
141	19.9	BIII	B	BIII/BIV	BIII/BIV
143	25.3	BIII	Negative	Negative	BIII
164	20.9	AII	AII	AII	AII
165	24.1	BIII	B	Negative	BIII
170	28.7	BIII/BIV	Negative	Negative	BIII/BIV
172	25.6	AII	B	Negative	AII+B
173	29.5	BIV	Negative	Negative	BIV
175	23.7	BIII/BIV	B	Negative	BIII/BIV
176	28.5	BIII/BIV	B	Negative	BIII/BIV
178	27.1	BIV	Negative	Negative	BIV
179	25.9	BIII/BIV	Negative	Negative	BIII/BIV
180	22.7	BIII/BIV	B	AII	AII+BIII/BIV
183	26.5	Negative	B	Negative	B
186	26.6	BIII/BIV	Negative	BIII	BIII/BIV
187	25.2	BIII	Negative	BIII	BIII
190	19.8	BIII/BIV	B	BIII	BIII/BIV
191	30.7	Negative	B	Negative	B
194	20.0	BIII/BIV	B	BIII	BIII/BIV
195	23.8	BIII/BIV	B	BIII	BIII/BIV
196	26.4	Negative	Negative	BIII	BIII
203	27.1	BIII/BIV	B	Negative	BIII/BIV
206	22.9	BIII/BIV	B	BIII	BIII/BIV

^1^ Symptomatic child, *bg*: ß-giardin, *gdh*: glutamate dehydrogenase, qPCR: real-time PCR, *tpi*: triose phosphate isomerase.

**Table 3 pathogens-10-00255-t003:** Diversity, frequency, and main molecular features of *Cryptosporidium* spp. sequences at the *ssu* rRNA locus in infected symptomatic and asymptomatic children in Zambézia province, Mozambique. GenBank accession numbers are provided.

Species	No. of Isolates	Reference Sequence	Stretch	Single Nucleotide Polymorphisms	GenBank ID
*C. hominis*	2	AF108865	608–979	None	MW563962
	1	AF108865	540–952	C607T, 697Del_T ^1^	MW563963
	1	AF108865	622–956	A808R, G905R	MW563964
*C. parvum*	1	AF112571	573–997	A646G, T649G, 686_689DelTAAT ^1^, A691T, C795Y, A892R	MW563965
	1	AF112571	630–997	A646G, T649G, 686_689DelTAAT ^1^, A691T, C795Y, T837W, A892R	MW563966 ^2^
	1	AF112571	539–1,025	A646G, T649G, 686_690DelTAATT ^1^, A691T, A892G	MW563967
	1	AF112571	655–985	Unknown ^3^	–
*C. felis*	3	AF108862	631–980	661InsT ^4^, T670A, 700DelT ^1^	MW563968
	1		648–987	661InsT, T670A,700DelT ^1^, 824InsT ^4^	MW563969
*C. viatorum*	1	KX174309	290–762	None	MW563970

^1^ Nucleotide(s) deletion. ^2^ Symptomatic child. ^3^ Sequence of insufficient quality to accurately determine the presence of potential single nucleotide polymorphisms. ^4^ Nucleotide insertion.

## Data Availability

All relevant data are within the paper and its Appendix A.

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
