# Peer review of "Molecular Diversity of Giardia duodenalis, Cryptosporidium spp., and Blastocystis sp. in Symptomatic and Asymptomatic Schoolchildren in Zambézia Province (Mozambique)"

_pathogens, 2021, doi:10.3390/pathogens10030255_

Round 1

Reviewer 1 Report

The manuscript by Muadica and collaborators is a thorough genetic and molecular characterization of the epidemiology of parasitic species causing diarrhoea in young children. The analysis is sound, the authors present a well structured report of their activities as well a well detailed methods section with all the protocols and primers sequences used.

This is highly appreciable in present times where authors tend to refer to the literature in order to give less than the minimum amount of practical tips.

This study is particularly interesting in that it presents the data of infection on a large panel of asymptomatic children in schools wide apart from each other, therefore sampling a large region in Mozambique.

The results are well explained and clearly reported, however there is no mention whether the prevalence of clustered SNPs found by the authors is associated to a particular set or a particular school.

Moreover from their Table 1, it seems that schools 17, 18, 2, and 13 have the highest prevalence of co-morbidity for Giardia spp. and Blasocyst spp. Nevertheless the children seems to be asymptomatic: the question arises whether the comorbidity is between hilghly mutated species or not. Are those schools in the same district or not? Have the children access to proper WASH settings or not?

All of those could be at least mentioned in the discussion, since it could help improving the protocols for disease control.

Minor points:

references 24 and 27 are now published, please update the link.

the italics for the species names of the parasites is missing in lines: 100, 101, 102, 110, 139, 159, 162, 169, 182, 192, 198, 202, 203, 204, 206, 209, paragraph 4.4, 4.5 and 4.6

The accession numbers of the deposited sequences are not cited -> please add them.

The footnotes no. 3-4 in Table 3 do not match with the upper-script c and d in the table text -> please revise

Author Response

Referee #1

The manuscript by Muadica and collaborators is a thorough genetic and molecular characterization of the epidemiology of parasitic species causing diarrhoea in young children. The analysis is sound, the authors present a well-structured report of their activities as well a well detailed methods section with all the protocols and primers sequences used. This is highly appreciable in present times where authors tend to refer to the literature in order to give less than the minimum amount of practical tips. This study is particularly interesting in that it presents the data of infection on a large panel of asymptomatic children in schools wide apart from each other, therefore sampling a large region in Mozambique.

We thank Referee #1 for his/her favourable preliminary comments on our manuscript.

  1. The results are well explained and clearly reported, however there is no mention whether the prevalence of clustered SNPs found by the authors is associated to a particular set or a particular school.

Response: No, we did not find evidence of association between a specific genetic profile and geographical location or presence/absence of symptoms (e.g. diarhroea). This result is in agreement with those previously published in the literature in similar surveys conducted in low-income countries including Mozambique. This is commented in current lines 290-291 and 300-302 of the Discussion section.

  1. Moreover from their Table 1, it seems that schools 17, 18, 2, and 13 have the highest prevalence of co-morbidity for Giardia and Blastocystis spp. Nevertheless the children seems to be asymptomatic: the question arises whether the comorbidity is between hilghly mutated species or not. Are those schools in the same district or not? Have the children access to proper WASH settings or not?

Response: regarding the predominance of G. duodenalis in apparently healthy children, please note that this result was highly expected, as G. duodenalis infections have been more prevalently reported in asymptomatic than in symptomatic children living in African Sub-Saharan countries including Uganda (see Ankarklev et al. PLoS Negl Trop Dis. 2012;6:2–9), Zambia (see Tembo et al. Food Waterborne Parasitol. 2020;19: e00072), Malawi (see Fan et la. Epidemiol Infect. 2019;147) and Ethiopia (see Damitie et al. Infect Dis Poverty. 2018;7: 1–10), among others. This issue has been thoroughly discussed in a recent paper by our research group (see Messa et al. PLoS Negl Trop Dis 2021, 15, e0008987). This paper is mentioned and commented several times in the present survey.

Regarding co-morbidity between G. duodenalis and Blastocystis, please note that indeed this association is not statistically significant (see results in Table 3 of Muadica et al. Clin Microbiol Infect 2020, in press, doi: 10.1016/j.cmi.2020.05.031). In practical terms this means that there is no point in investigating whether a given genetic profile is more frequently found in co-infected children that those with single infections. We strongly believe that the extraordinary genetic diversity found within G. duodenalis is the direct consequence of the hyperendemicity present in this epidemiological scenario. Please note that this point is already mentioned in the Conclusions section.

Regarding WASH, most of the schoolchildren populations investigated in the present survey came from rural areas with little or no access to safe drinking water and sanitation. This point has been now highlighted in the Conclusion section of the manuscript.

  1. All of those could be at least mentioned in the discussion, since it could help improving the protocols for disease control.

Response: some of the points raised by Referee #1 has been now added in the Discussion and Conclusion sections of the manuscript as recommended.

Minor points:

  1. references 24 and 27 are now published, please update the link.

Response: Please note that reference [24] corresponds to a manuscript that is being currently submitted to the special issue “Diagnosis, Epidemiology and Transmission Dynamics of Cryptosporidium spp. and Giardia duodenalis” in Pathogens. A PDF copy of this manuscript is available under request.

Reference [27] is still in press. The doi and PMID numbers of this manuscript are as follow: doi: 10.1016/j.cmi.2020.05.031 and PMID: 32505583.

  1. the italics for the species names of the parasites is missing in lines: 100, 101, 102, 110, 139, 159, 162, 169, 182, 192, 198, 202, 203, 204, 206, 209, paragraph 4.4, 4.5 and 4.6

Response: Corrected. Italicised names of species and gene abbreviations have been re-checked through the whole manuscript.

  1. The accession numbers of the deposited sequences are not cited -> please add them.

Response: Pending GenBank accession numbers for Cryptosporidium gp60 sequences and Blastocystis ssu rRNA sequences have been added as per requested.

  1. The footnotes no. 3-4 in Table 3 do not match with the upper-script c and d in the table text -> please revise.

Response: Thanks for spotting this mistake. Corrected as per requested.

Reviewer 2 Report

This work is very valuable and contributes to understanding the distribution and prevalence of three important intestinal protists associated with the occurrence of diarrhea in a large, carried out on a large, representative group of children in Mozambique. The research was carried out using advanced molecular techniques, which is all the more valuable as research on the presence of these pathogens in Africa is very sparse and usually performed using microscopic methods, which are less sensitive, and therefore most likely understated. The use of advanced molecular methods made it possible to analyze the genetic diversity of these protists in the studied population.

The obtained results fill the gap in knowledge about the distribution of genotypes  of G. duodenalis, Cryptosporidium spp. and Blastocystis sp. in different geographical regions.

I present my comments on the manuscript below:

More serious comments:

Lines 71-72: Similarly, 22 distinct subtypes (ST) have been identified within Blastocystis sp. to date, 72 of which ST1–9 and ST12 have been reported in humans [16].

Comment: The authors of the cited publication do not write that there are 22 subtypes of Blastocystis, but write that there are 17 certain, confirmed subtypes, and among the others (18-26) they are either not valid or require further research to confirm that they are valid.

Lines 252-253: Neither mixed infection involving different STs of the parasite nor infections caused by animal-specific ST10-ST17, ST21 or ST23-26 were identified.

Comment: I suggest re-editing this sentence (do not list the subtypes that have not been identified, but only those that have been identified), for instance: Neither mixed infection involving different STs of the parasite nor infections caused by any of remaining subtypes were identified.

Similar, line 362 should be re-edited.

If the Authors do not agree with above suggestions, they should refer to other publications on Blastocystis subtypes, not [16].

It is the Authors' right to request that the deposited sequences be not available until the publication is out (this is the case with sequences of Cryptosporidium spp. obtained in this study). But it seems that sequences of G. duodenalis are not deposited in GenBank (numbers given in ms but they are not deposited in GenBank), as well as Blastocystis (XXXX instead of numbers in ms). Please fill in these gaps.

Table S3 - in last column: „GenBank ID” – accession numbers of sequences not provided

Minor remarks:

Species / genera names, as well as the names of genes and loci should be italicized: incorrect spelling appears many times in Results, caption under Figure 1, Figure 2, Materials and Methods.

Full species name should be given only the first time, and then abbreviated, eg Giardia duodenalis / G. duodenalis. The whole parasites names are given in many places of the manuscript as well as in descriptions of the Figures and Tables.

Line 244 and 327: should be ssu rRNA instead of ssu rDNA

Author Response

Reviewer #2

This work is very valuable and contributes to understanding the distribution and prevalence of three important intestinal protists associated with the occurrence of diarrhea in a large, carried out on a large, representative group of children in Mozambique. The research was carried out using advanced molecular techniques, which is all the more valuable as research on the presence of these pathogens in Africa is very sparse and usually performed using microscopic methods, which are less sensitive, and therefore most likely understated. The use of advanced molecular methods made it possible to analyze the genetic diversity of these protists in the studied population. The obtained results fill the gap in knowledge about the distribution of genotypes of G. duodenalis, Cryptosporidium spp. and Blastocystis sp. in different geographical regions. I present my comments on the manuscript below:

We thank Referee #1 for his/her favourable preliminary comments on our manuscript.

More serious comments:

  1. Lines 71-72: Similarly, 22 distinct subtypes (ST) have been identified within Blastocystis to date, 72 of which ST1–9 and ST12 have been reported in humans [16].

Comment: The authors of the cited publication do not write that there are 22 subtypes of Blastocystis, but write that there are 17 certain, confirmed subtypes, and among the others (18-26) they are either not valid or require further research to confirm that they are valid.

Response: Please note that the 22 STs mentioned in the paper include only those accepted as valid (ST1-17) or in need of further confirmation because of limited length of the available ssu rRNA sequences (ST21, ST23-26). Rejected STs including ST18-ST20 and ST22 were not considered, so the final number of valid/proposed STs is indeed 22. This issue has been now clarified in current line 72 of the manuscript.

  1. Lines 252-253: Neither mixed infection involving different STs of the parasite nor infections caused by animal-specific ST10-ST17, ST21 or ST23-26 were identified.

Comment: I suggest re-editing this sentence (do not list the subtypes that have not been identified, but only those that have been identified), for instance: Neither mixed infection involving different STs of the parasite nor infections caused by any of remaining subtypes were identified.

Response: We disagree with Reviewer #2 comment. Please note that the point here is confirming the absence of Blastocystis STs known to be animal-specific to date (or, in other words, of unknown or limited zoonotic potential). We believe that these STs should be clearly identified in the text to avoid misleading interpretations by non-specialised readers.

  1. Similar, line 362 should be re-edited.

Response: Please see our answer to comment #2 above.

  1. If the Authors do not agree with above suggestions, they should refer to other publications on Blastocystis subtypes, not [16].

Response: Please see our answers to comments #2 and #3. Please also note that reference [16] provides the most widely accepted taxonomic classification of Blastocystis sp. STs currently available, allowing direct comparison of sequence data from different studies in a simple an effective way.

  1. It is the Authors' right to request that the deposited sequences be not available until the publication is out (this is the case with sequences of Cryptosporidium obtained in this study). But it seems that sequences of G. duodenalis are not deposited in GenBank (numbers given in ms but they are not deposited in GenBank), as well as Blastocystis (XXXX instead of numbers in ms). Please fill in these gaps.

Response: Pending GenBank accession numbers for Cryptosporidium gp60 sequences and Blastocystis ssu rRNA sequences have been added as per requested. Please note that ssu rRNA sequences are immediately available on GenBank after submission, whereas protein coding sequences (such as those for the gdh, bg, tpi, and gp60 loci) are only available after manuscript´s acceptance. Proof of sequence submission to GenBank database is available under request.

  1. Table S3 - in last column: „GenBank ID” – accession numbers of sequences not provided

Response: GenBank accession number for tpi sequences shown in Table S3 have been now provided.

Minor remarks:

  1. Species / genera names, as well as the names of genes and loci should be italicized: incorrect spelling appears many times in Results, caption under Figure 1, Figure 2, Materials and Methods.

Response: Corrected. Italicised names of species and gene abbreviations have been re-checked through the whole manuscript.

  1. Full species name should be given only the first time, and then abbreviated, eg Giardia duodenalis / duodenalis. The whole parasites names are given in many places of the manuscript as well as in descriptions of the Figures and Tables.

Response: Corrected. Please note that full names have been maintained at the beginning of a new sentence, and also in Figure legends and Table titles to meet the editing publication standards of Pathogens.

  1. Line 244 and 327: should be ssu rRNA instead of ssu rDNA

Response: Corrected as per requested.